# Contested Development Paths and Rural Communities: Sustainable Energy or Sustainable Tourism in Iceland?

**Anna Dóra Sæþórsdóttir [1],\* and C. Michael Hall [2]** 

1   Faculty of Life and Environmental Sciences, University of Iceland, 101 Reykjavik, Iceland
2   Department of Management, Marketing and Entrepreneurship, University of Canterbury, Christchurch 8140 2, New Zealand
\*   Correspondence: annadora@hi.is; Tel.: +354-5254-287

**Abstract:** The Icelandic economy has transitioned from being dependent on fishing and agriculture to having tourism and refined aluminum as its main exports. Nevertheless, the new main industries still rely on the country's natural resources, as the power intensive industry uses energy from rivers and geothermal areas whereas tourism uses the natural landscape, where geysers, waterfalls and thermal pools are part of the attraction to visitors. Although both industries claim to contribute to sustainability they utilize the same resources, and land-use conflicts can be expected, illustrating the contestation that can occur between different visions and understandings of sustainability. This paper focuses on the attitudes of Icelandic tourism operators towards power production and proposed power plants using data from questionnaires and face-to-face interviews. Results show that the majority of Icelandic tourism operators assume further power utilization would be in conflict with nature-based tourism, and they are generally negative towards all types of renewable energy development and power plant infrastructure. Respondents are most negative towards transmission lines, reservoirs and hydro power plants in the country's interior Highlands. About 40% of the respondents perceive that existing power plants have negatively affected tourism, while a similar proportion think they had no impact. According to the respondents, the two industries could co-exist with improved spatial planning, management and inter-sectoral cooperation.

**Keywords:** land use conflicts; tourism industry; nature-based tourism; sustainable power production; renewable energy development

## 1. Introduction

Many sparsely populated areas in the northern high latitudes have utilized energy production as a key means of economic development. Although primarily focused on hydro-electric development other forms of energy, such as thermal and nuclear energy and oil and gas resources, and more recently wind power, have also proven significant [1–4]. Initially, such developments were often encouraged by governments because of the relatively low population levels, the perception that the land had little other economic value, and that there were few other economic and regional development alternatives [2,5,6]. However, the emergence of the modern conservation movement, which strongly embraces the conservation of areas with high natural and wilderness values, means that nature-based tourism has become a significant economic trajectory for sparsely populated areas [7–12]. Although sharing some common infrastructure, such as the importance of transport access, energy developments and nature-based tourism have different patterns of resource use in both space and time [13–16]. At the same time energy-related developments can also become significant tourist attractions in their own

right and provides a non-competing for of use [17]. The relationships between tourism and energy development in areas with high perceived natural values therefore provide a potentially contested policy environment for selecting different sustainable development paths and raises fundamental policy and resource management questions as to the extent to which tourism and energy development are compatible forms of economic and resource development.

It has long been recognized that there are significant issues surrounding the different communities and stakeholders that are impacted by energy developments and their capacity to influence change in peripheral areas [18,19]. These issues become more complicated when there is the potential to take different development paths in which one trajectory would affect, or even negate, the potential of other trajectories [13,20,21]. Gaining and improved understanding of choices between different development paths is also important with respect to sustainability as they can represent different understandings of what constitutes sustainable development, renewable energies or sustainable tourism [22–25]. In the Icelandic context, for example, renewable energy developments, nature-based tourism, and visits to energy infrastructure, such as hydro-electric developments, have all been framed at various times as contributing to sustainable development. This common though contested framing of sustainability arguably has been at the contested core of conservation thinking for well over a century [26,27]. The advocacy of renewable energy projects being associated with the 'wise-use' or 'progressive conservation' movement, while limiting such developments and advocating for economic development through tourism is associated with the Romantic conservationist of the early wilderness movement [28,29]. Indeed, in many peripheral areas of developed countries there is a long history of renewable energy developments, such as those surrounding hydro-electric developments, being opposed by groups that favor development paths that are dependent on wilderness tourism [2,30], or agriculture [31]. Despite a usually high level of support for renewable energy in general, attitudes towards specific projects can be negative among some communities and stakeholders in the policy process, focusing both on different economic visions, as well as understandings of valued landscapes [32–37].

Some studies [38,39] have indicated that renewable energy infrastructure reduces the attractiveness of nature-based tourism destinations. According to Fredman and Tyrväinen [40] the nature based tourism sector generally owes its business to the perceived naturalness of the landscape. Consequently, conflicts can be expected between renewable power production and the nature-based tourism sector. Furthermore, these conflicts are often greater in high-quality natural areas [41] than in areas where there are already industrial plant and infrastructure [42].

Some renewable energy infrastructure, such as wind turbines are immense constructions and can therefore have a significant impact on the perceived naturalness of a landscape. Several studies [38,43–46] point out that the main reason for the disapproval of wind farms is the apprehension that perceived quality of the landscape will diminish. Several studies [47,48] point out that travelers sometime stop visiting a tourist destination after wind farm construction. On the other hand, a new segment of tourists sometimes starts to visit an area after the development of power plant, as energy infrastructure can be an attraction for some, while associated roading can make access easier [43,49]. According to Frantál and Urbánková [17] such 'energy tourism' is growing as a product. In addition, economic reasons can also be a factor that affect the attitudes towards power plant development, e.g., if economic benefits are to be expected people might be willing to sacrifice some of the perceived quality of the nature [50–52].

Iceland has the benefits of growing tourism industry as well as various options for producing renewable energy. So far, several hydro-electric and geothermal power plants have been built and more are under consideration. In addition, the development of the first wind farms in the country are now being discussed. However, studies among tourists at natural area destinations in Iceland show that tourists are negative towards proposed renewable power plant development in natural areas [53–56]. On the other hand, a study by Sæþórsdóttir and Hall [57] shows that where a hydro

power plant infrastructure has been developed tourists' perception of the naturalness of the area in question has not been severely affected.

When new power plants are built in natural areas new roads are built and improvements made to existing roading. Therefore, new areas become more accessible, which, sometimes lead to increased number of visitors although the experience of the place and the market group that is attracted might be different from what existed previously [57,58]. Consequently, road construction and accessibility have a major effect on how a tourist destination develops, its characteristics, what type of tourists visit the area and to what extent the area is visited [59,60].

Based on surveys and interviews with tourism operators the study presents an overall qualitative analysis of stakeholder perceptions. The aim of this paper is to examine:

(1)　How tourism operators perceive the ideas of further development of power plants in Icelandic nature.

(2)　What impact tourism operators think proposed power plants have on the tourism industry in Iceland.

(3)　If tourism operators see new market opportunities or a loss of opportunities due to power plant development.

(4)　If tourism operators perceive that tourism and power production can coexist.

(5)　If tourism operators perceive tourism or power plants as a better alternative for regional development, or if conflicts foreseeable.

The research fills an existing gap in knowledge in this field, that is the view of the tourism industry on energy developments. That knowledge is of vital importance as the opinion, beliefs and perceptions of the sector influences their actual behavior. This does not mean that the business will be correct with respect to what will happen to the operation or to tourism as a whole if a power plant is constructed, but beliefs do influence actions and decision-making [61]. If, for example, an operator in a rural area does not think that their market could coexist with a power plant, especially one planned to be built in the near future, then they might not spend time or money marketing a tour into the area or may even operate elsewhere. Alternatively, the development of a power plant may be regarded as a potential opportunity to develop new product offerings and develop new markets given increased accessibility and new attractions [17].

In Iceland such information is of special importance as the tourism industry has become the largest export sector [62], while the government also seeks to identify the best economic and environmental conservation strategies for areas that are currently perceived as having high natural values [58,63,64]. The paper stresses furthermore the need to understand the selection of development paths for peripheral locations as a result of horizontal and vertical sets of relations between actors within multi-level governance and decision-making structures. In such instances the capacities of individual communities and policy actors to influence economic outcomes is both constrained and enabled by these broader structures and in many cases the key decisions that affect sustainable tourism and economic trajectories are made elsewhere. This appears particularly to be the case where energy resource decisions are concerned. In addition, the paper discusses some of the inherent difficulties of managing the complex processes affecting new development paths in sparsely populated countries and regions where development options may be limited.

## 2. Energy and/or Tourism in Iceland

### 2.1. The Changing Economy of Iceland

Iceland is an island in the North Atlantic Ocean, between 63° and 66° northern latitudes. Its area is 103.000 km$^2$, almost 60% of which lies at altitudes above 400 m and 24% lies below 200 m. The population of the country is about 338,000 and almost all settlement is below 200 m. About 64% of the population

lives in the capital region, and the rest in towns and villages scattered along the coast and on farms on plateaus along the coast and in valleys that penetrate the country [65] (Table 1).

**Table 1.** The population of the regions of Iceland in 2019.

| Region | Population | Population Share (%) |
|---|---|---|
| Capital region | 228,231 | 63.9 |
| Southwest | 27,114 | 7.6 |
| West | 16,765 | 4.7 |
| Westfjords | 6961 | 1.9 |
| Northwest | 7071 | 2.0 |
| Northeast | 30,452 | 8.5 |
| East | 13,052 | 3.7 |
| South | 27,345 | 7.7 |
| Total | 356,991 | 100.0 |

Source: Statistics Iceland [65].

Iceland has a resource-based economy. Grassland for sheep farming, fish in the ocean, the energy sector—driven by hydro and geothermal power—and a wild landscape that attracts international tourists. Agriculture was the major sector in the nineteenth and early twentieth centuries, but declined after World War II. Traditionally, seafood was the dominant export sector, but its share has declined and was about 18% in 2018 [62]. This changing economy has led to outmigration from rural areas to the capital area. Depopulation and decline has therefore characterized most of the traditional farming and many of the fishing communities. Both sectors have become more technological oriented and employ less labor. In addition, fish cutting is increasingly undertaken by immigrant workers. Since 1990 fishing quotas have been transferable between regions. This has led to fishing rights being transferred between regions leaving many communities without any fish to process and fishing communities very vulnerable. In addition, being close to main markets and having access to good transportation is increasingly important, especially for export of fresh fish [66].

The export share of aluminum products of the total export was about 17% in 2018 [62]. The first aluminum smelter in Iceland was built in the mid-1960s, since then six more international energy intensive factories have been built and they use about 77% of the total energy produced in Iceland [67]. So far seven large (> 90MW) hydro power plants have been built in the Central Highlands, the unpopulated wilderness interior of the country. Six geothermal power plants have been built and the seventh is under construction. These are all in the lowlands.

Most of the energy intensive industries are located close to the capital region in the south-west corner of the country. One of the exceptions is the aluminum smelter Fjarðaál located in Reyðarfjörður, a fishing and trading port in the East, and owned by the American multinational company Alcoa. For the smelter's energy needs the Kárahnjúkar hydro power plant was built in the nearby Highlands. This is by far the largest hydro-plant in the country. When the smelter began production in 2007 it used about 40% of all electrical power produced in Iceland. The argument for the development of this megaproject was that it would create new jobs and bring welfare benefits to a region which had been seeking to counteract depopulation [68]. In 1998 there were only 683 persons living in the town of Reyðarfjörður so human resources for the aluminum smelter came from other nearby fishing villages, mainly in the municipalities of Fjarðabyggð and Fljótsdalshérað, which is a service and agricultural area further inland. In order to bolster the labor market road improvements and a tunnel were financed by the national government. New jobs were indeed created. Some of them temporarily during the construction period, when 80% of those employed were foreign migrant workers, given that only 2% of the Icelandic workforce were unemployed at the time and only 1.2% in the East of Iceland [69]. Currently, about 544 people work in the factory [70] and about 350 others hold various related positions, hence almost 900 jobs have been created permanently. About 95% of the Alcoa staff live in the nearby communities, 62% in Fjarðabyggð and 30% in Fljótsdalshérað [70]. In the Kárahnjúkar hydro power plant itself there are 13 permanent employees [71].

The agreement regarding the construction of the Alcoa smelter was made in 2003. Following the construction of the power plant and the start of smelter operations the population increased, reaching a period of peak growth of over 18% annually in 2006 and 2007 in the Fjarðabyggð and Fljótsdalshérað municipalities (Figure 1). After construction was finished the population declined again, although it has now stabilized. Between 2003 and 2019 the population increased by 1871 or about 27%. The Fjarðabyggð municipality also receives about 600 m. ISK of income annually from the smelter [69] which in 2009 was about 15% of their operating revenue. Overall, about 36% of the company's export income remains in Iceland or about 81 billion IKR in 2017 [70], of which about 12% is salaries and purchased goods and services about 43%. In addition, Landsvirkjun, the national power company, pays about 87 m. ISK in real-estate tax to the Fljótsdalshreppur municipality, which is 58% of its annual income [72].

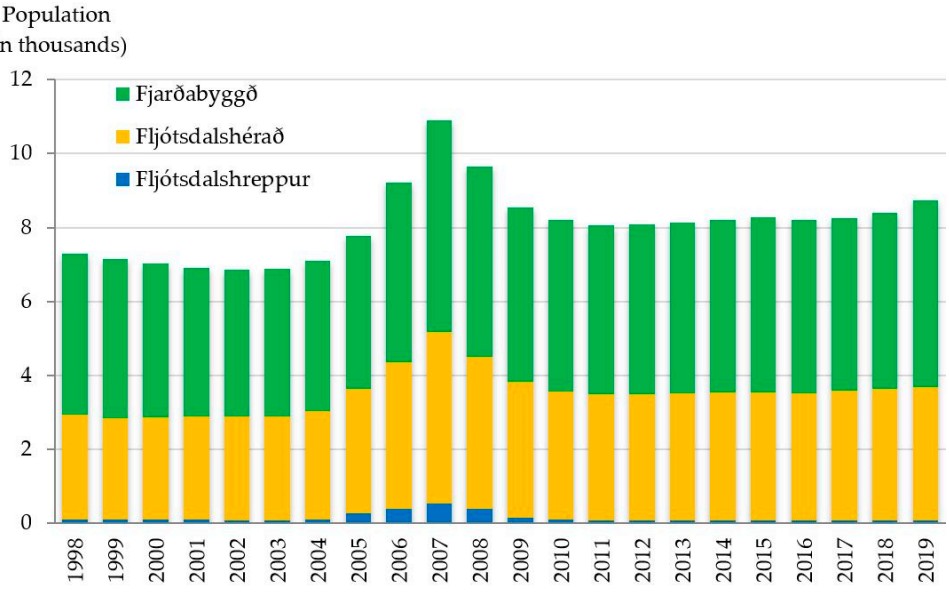

**Figure 1.** Number of inhabitants in the three municipalities most affected by the Kárahnjúkar power plant and Fjarðaál—Alcoa aluminum smelter. Analyzed from Statistics Iceland [65].

The tourist industry provided between 12% and 13% of total exports between 1995–2009. However, since then it has become a critical part of the Icelandic economy with its share of foreign exchange earnings has increased from 19% to 39% in 2010–2018 making it now by far the largest export sector [62]. This growth in export from tourism has mainly been the result of the increased number of international tourists that come to Iceland [73], which has increased on average by 13% a year, in the last 20 years. The increase has been particularly great since 2010, with an average annual growth of 22%. In 2017 as many as 2.2 million foreign tourists arrived in the country, which is over seven times more than the Icelandic population. About 98.7% of them arrived on flights at Keflavík International Airport, which is located at the south west corner of the country, about a 30-min drive from Reykjavík. Approximately 22,000 or around 1.0% of the total came on a boat through Seyðisfjörður on the east coast and about 6500, or 0.3% of the total, came on flights through Reykjavík Airport or Akureyri Airport [74]. In addition, about 145,000 cruise ship passengers came to the country in 2018, with an average annual increase of 9% from 2010 [75].

Tourism in Iceland has been very seasonal with a summer high season, but this has been changing. In 2017 around 35% of tourists arrived in the three summer months, June, July and August, while in 2010 50% came at this time. The number of visitors travelling in the 'low-season', i.e., the other nine months increased from 691,000 in 2010 to 1,417,000 in 2017, which is more than double [74]. Tourism has thereby begun to offer a variety of year-round employment, although significant spatial variation remains. During the summer months, 92% of international tourists come to Iceland because of nature

and 89% during the winter [76]. About 44% of those who come because of nature are fascinated by the fact that it is unspoiled, while about a quarter name either natural beauty, Geysir/geothermal areas, uniqueness and diversity or landscape and scenery [77].

About 28,000 jobs, or 14% of the total workforce is employed in tourism and 4370, or 2% in energy intensive industry, which is mostly aluminum production [78,79]. The tourism industry has been the main provider of new jobs since the financial crisis [80] and especially since 2010 there has been a 60% increase, while jobs in other sectors have increased by about 14% [79]. About 65% of jobs in the tourism industry are in the capital region and about 35% in the rural areas. In contrast, in the energy intensive industry 61% is in the rural areas and 39% in the capital region [79] (Figure 2).

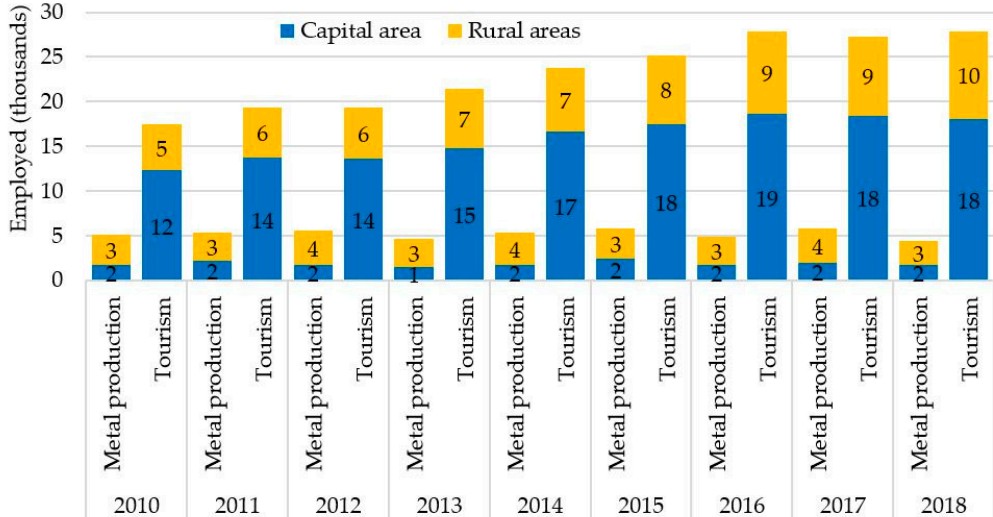

**Figure 2.** Employment in the tourism sector and the energy intensive industry. Analyzed from Institute of Economic Studies and Statistics Iceland [78,79].

About 70% of Iceland's gross domestic product (GDP) is attributed to the capital region and this figure increased about 5% from 2009–2013. In this same period population increase in the capital area was about 3% while the rest of the country had a 1% decrease. In the same time period, the Southwest and the Westfjords experienced a regional GDP decline of about 11–12%. The most growth was in the West, about 13%, this was most likely due to the development of energy intensive industry, other businesses and the public sector. However, data regarding population and salary does though not point to any significant increase there [66].

## 2.2. Land Rent from Common Pool Resources

Both energy intensive industry and tourism are natural resource based. With nature and wilderness as the main resource for the tourism industry in Iceland, some conflicts appear unavoidable between the two economic sectors. The visual impact of power plants in the Icelandic landscape is significant. Hydro power plants comprise dams, canals, reservoirs, and the power station housing the turbines and transformers although the power stations are often partly located below ground level. Hydro power plants also impact the neighboring environment and can affect water flows and riverine and estuarine vegetation. The geothermal power plants require large buildings for turbines and steam separators, the drill holes are noisy and emit steam and are connected to the main buildings by pipelines. The characteristics of the geothermal areas are affected by energy development and made less interesting to observe for tourists because of the presence of buildings and infrastructure and changes in geothermal activity. Power plants are also accompanied by transmission lines and their visual impact is massive, especially in wilderness areas, as the landscape is very barren and there are usually no trees to otherwise conceal the pylons. As of early 2019 only two experimental windmills have been built and are located at the edge of the Highlands, although more have been proposed [81]. In addition, some farmers have

built 2–3 windmills in the lowland. Research among travelers in the Highlands has shown that the majority of tourist are negative towards power plants in the destination where they are travelling and they are considered to have a negative effect on the wilderness experience [19,53,54]. The majority of tourists claim that wilderness is an important part of the appeal of the Highlands [54,82]. Nevertheless, it has been pointed out [83] that vision is socially constructed. All vegetated land in Iceland has through the centuries been affected by overgrazing and the felling of wood for smelting, and this effect extended throughout the Highlands [84,85]. The only areas that are relatively pristine were the ice caps and the higher areas of the Highlands, where there was no vegetation due to physical conditions. Nevertheless, despite the substantial changes that have occurred to the Icelandic environment, the promotion of Iceland as a tourist destination continues to be based on notions of wilderness and high levels of naturalness [83].

Municipalities in which power plants are built receive a property tax from the plant, which is where the power station itself is located but often not where the reservoir, dam, or transmission lines are. The municipalities where the transmission lines cross the land do not receive property tax [72]. The spatial characteristics of renewable energy production, transmission and consumption in Iceland means that energy is often used 'elsewhere', i.e., the energy intensive industries are not necessarily located in the municipality where energy production occurs and where environmental impacts occur. This situation can have a significant impact on the creation of permanent employment after the construction phase is complete. Although plants do generate income for municipalities there are ever more questions as to the relative value of such returns from the municipalities themselves, as well as their long-term contributions to employment generation [72]. The CEO of Landsvirkjun, the national power company of Iceland, even suggests that there are five stakeholders who could claim rent from the natural resource: the owner of the land, the power producer, the buyer of the energy, the local community and the nation as a whole. He furthermore argues that in order to maximize the value of the natural resources it is important that the distribution of the rent between the different stakeholders is 'fair' [86].

While tourism is perceived as creating jobs in many of the rural municipalities a major problem is that often there are so few people living there permanently that they need to employ people from outside the region (mostly from Reykjavík or from the EU) to work in the tourism industry during the high season. Significantly, such short-term contract employees are not registered for paying tax in the municipality but where they have a permanent address [87]. Therefore, only very limited local tax stays in the community [87,88]. The VAT from the tourism industry on the other hand goes directly to the state but not to the local government. As Karlsson et al. [87] point out, that is quite an uneven system as some municipalities are kind of a 'travel through areas' which carry costs due to tourist visitation for example due to sewage and garbage but they don't receive any income as tourist don't spend any money there.

The situation for tourism has also become further complicated by the lack of available housing for employees and higher rents for residents, especially given that much previously available housing is now being rented out as short-term accommodation for tourists, often through Airbnb [88]. According to the Housing Financing Fund [89], short-term accommodation contributed to an estimated 5–9% increase in rental prices in Iceland between 2015–2017. To counter this the government passed a law which limits the number of days in each year, an owner can short-term lease their apartment [90] and some municipalities have even elected not to grant any new home sharing permits [91].

It needs to be acknowledged that both industries have discussed the need to change the taxation system in order to generate greater returns to the municipalities [72]. Undoubtedly, this would then influence the attitudes among the inhabitants towards the various industries. Nevertheless, the various resource demands of the two sectors clearly create challenges for the selection of economic strategies and the consequent potential for lock-in of an economic trajectory that may prove disadvantageous to the region in the long-run.

## 3. Background and Methods

### 3.1. The Icelandic Master Plan for Nature Protection and Energy Utilization

In order to try and generate greater consensus on the use of energy resources in Iceland a government project called The Icelandic Master Plan for Nature Protection and Energy Utilization (Áætlun um vernd og orkunýtingu landsvæða, 'rammaáætlun') commenced in Iceland in 1999. Under this project all potential power plant projects are evaluated and ranked with respect to their economic and environmental impact. The objective of the master planning project was to integrate utilization and conservation policies and improve the overall planning process, potentially leading to a greater consensus on the harnessing or protection of natural resources. The Master Plan project has, so far, been split into four phases: Phase 1, 1999–2003; Phase 2, 2004–2010; Phase 3, 2013–2017 and the current Phase 4 which started in 2017. The project is led by a steering committee, but most of the work is carried out by four groups of specialists. One of the groups (workgroup 2) consists of nine experts and evaluated the effects of power plant development on recreation and tourism as well as other economic use such as grazing and fishing. In Phase 3 a total of 84 proposed power plant projects were forwarded by the National Energy Authority for evaluation. Due to limited time and budget the steering committee decided that main focus should be on 26 proposals: seven geothermal, one of which is in the Highlands, and 17 hydro power, 11 being in the Highlands. In addition, for the first time, two windfarms are being evaluated, both at the edge of the Highlands (Figure 3). The data introduced in this paper was gathered for workgroup 2 and was a part of a background material for their evaluation [92–96].

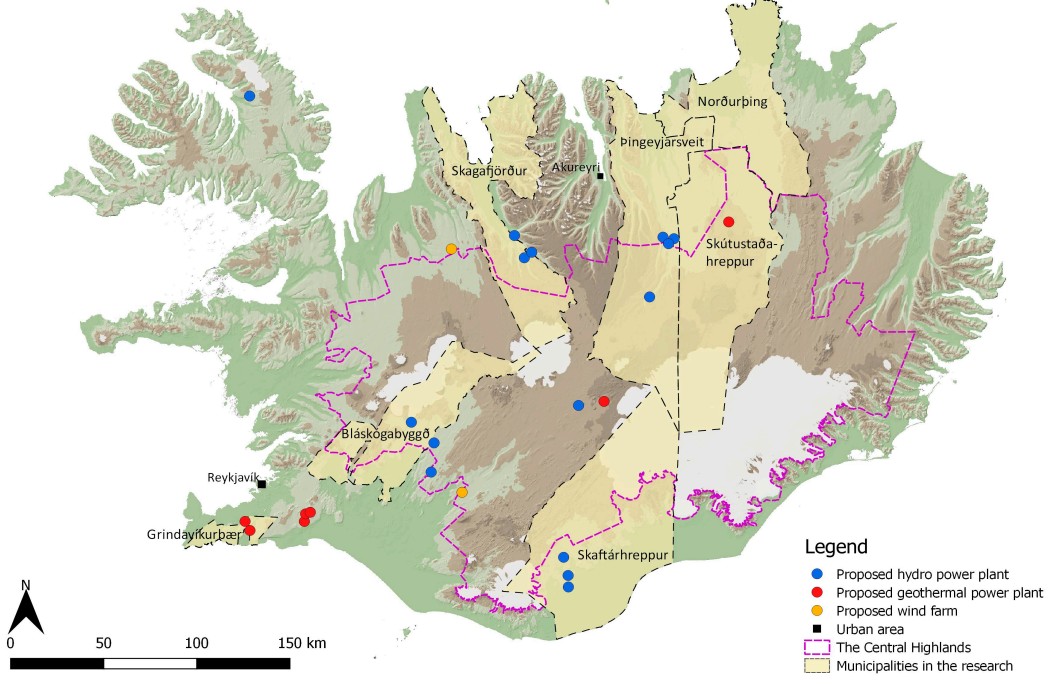

**Figure 3.** Proposed power plants in 3rd phase of the Master Plan and the municipalities where the interviews were conducted.

### 3.2. On-line Questionnaire Survey

A survey was developed to investigate the response of tourism industry members to the proposal evaluated in the Icelandic master plan with respect to power plants and energy infrastructure. Questions were developed in light of some of the existing literature on responses to energy infrastructure in areas with high perceived nature values [38,39] as well as previous research on the tourism industry in Iceland [92–96].

A link to an on-line questionnaire survey was sent with an e-mail to all tourism operators with licenses from the Icelandic Tourist Board, a total of 986. It was made with QuestionPro software tool (www.questionpro.com) and was open from 24 November until 1 December 2015. A reminder was sent 30th of November and time extended until 3rd of December.

It contained 21 question that can be grouped into:

- Their type of business and location of where they take tourists/run their business.

    – What kind of tourism services do you provide?
    – Where does most of your business take place?
    – How many employees work at the company?
    – How many years have you been in business?

- The effect of existing power plants on the tourism industry.

    – Have the existing power plants had an impact on your business or the way you run it? Has it been good or bad?

- Attitudes towards the various types of power plants (hydro, geothermal wind) and related structures as well as their location (Highland versus lowland) and their further development.

    – Please state how positive or negative your attitude is to the following:

        ○ Hydro power plants in the Highlands
        ○ Hydro power plants in the lowlands etc.
        ○ Further development of hydro power plants in the Highlands
        ○ Further development of hydro power plants in the lowlands etc.

    The replies were based on a five-point Likert scale, i.e., 1 = very negative, 2 = somewhat negative, 3 = neutral, 4 = somewhat positive, 5 = very positive.

- Attitudes towards the various 26 power plant proposals. Here the respondents could open a link at the webpage of the National Energy Authority with a brief description of each of the 26 proposals.

    – How would you rate the following power plant proposal (on the scale 0 = very bad – 10 = very good) regarding how good or bad you think it has on i) the tourism industry on your company and in Iceland).

In total 355 opened the survey, 259 started and 156 finished it, corresponding to a 15.8% response rate. It took those who finished on average about 14 min to fill it out. In the analyses both descriptive statistics were used as well as in the Likert scale questions the means were calculated. In order to discover whether there was a statistically significant difference between the tourism operators' preferences of the various forms of power production in the Highlands or lowlands, independent t-tests were used. Additionally, to compare tourism operators' evaluations of the effect of each of the 26 power plant proposals in the Master Plan project on the tourism industry and on their company paired t-tests were used. In the following analyses, a significance level of 0.05 is used, i.e., if $p < 0.05$ it is concluded that statistically significant differences exist.

### 3.3. Semi-Structured interviews

In order to add depth to the information given by the questionnaire survey and to capture the complexity of the subject, face-to-face interviews were conducted with 65 tourism operators. A purposive strategy sample was used in selecting the interviewees. Most (42) were in the municipalities

that neighbor where the proposed power plants are located, that is in Skagafjörður, Þingeyjarsveit, Skaftárhreppur, Bláskógarbyggð, Grindavíkurbær (Figure 3 and Table 2). Seventeen were located in Reykjavík and six in Akureyri (the second largest urban area after the Capital area) but organize tours into the areas or run accommodation (e.g., mountain huts) near where there are proposals for power plant developments. Most of the companies in the rural communities organize tours in the areas where proposed power plants would be located, but some also offer accommodation. Some companies offer both accommodation and some other kind of an activity, such as hiking, horseback-riding or jeep tours.

The interviews were semi-structured and contained mostly open-ended questions grouped around the following themes and questions:

- Current utilization of the area (quantity of use and what type of activities/tourists).

    – What kind of business do you run in the area?
    – How many visitors (in your tours, at you place at each time/ in a year)?
    – What do they do while they are here?

- The area's attraction and its uniqueness as a tourist destination.

    – What kind of tourists do you get and what are their demands?
    – What places are they looking at, what are the most important ones, what is the attraction for travellers?
    – What makes this place special as a destination for travellers?

- Vision and future possibilities for tourism in the area.

    – What possibilities do you see (in the area) for the future?
    – How do you see the tourism industry developing in the next years?
    – Could the area be used more by the tourism industry? How?
    – Could more travellers come into the area? Can the area tolerate more travellers? Why?
    – What kind of infrastructure is suitable in this area?
    – What do you think future travellers would prefer to have here regarding access and infrastructure (accommodation, restaurants/catering, activities?)

- Attitudes towards power plant proposals in the operating area and possible influence on tourism.

    – Are you familiar with the power plant proposals? (explain shortly if they are not)
    – What is your opinion on each of them?
    – Would the power plants have impact on you/your company? If yes, how?
    – What impact would these power plants have on tourism? (your own business, others, travelers experience)?
    – How do you think tourism would develop with the power plants?
    – How do you think tourism would develop without the power plants?
    – What do you prefer—why, argue for it?

- Tourism and/or power production as a solution/economic trajectory for the municipality and national interests.

    – Which do you consider the preferred alternative for Iceland: further development of tourism or energy production? What about for regional development within the municipality? Alternatively, can they coincide?

The interviews were conducted over the winter of 2016 and took place either in respondent's offices or homes and took from 15 min to one hour, with an average of about 35 min. Maps of the

proposed power plants where presented to the interviewees to encourage and focus discussion. The interviews were recorded and transcribed and analyzed according to the themes that emerged. The interviews reflect on individuals' version of 'truth' [97] and the analysis of the interviews is built on a standpoint that beliefs about 'nature' are not fixed ontological properties but a socially produced 'reality' [98,99]. The analysis was grounded on the identification and thematic classification of the participants perceptions and opinions [100]. Axial coding was then used to identify the connections between the key characteristic and associated conditions [101].

**Table 2.** Location and type of tourism business of the interviewees.

| | Reykjavík | Akureyri | Árnessýsla | Grindavík | Skaftárheppur | Skagafjörður | Þingeyjarsveit | Total |
|---|---|---|---|---|---|---|---|---|
| Accommodation | | | 3 | | 2 | 4 | 3 | 12 |
| Accommodation + tours | 1 | | 2 | | 3 | 8 | | 14 |
| Coach/jeep tours | 7 | 3 | | | | | 3 | 13 |
| Driving and hiking tours | 4 | | | | 1 | | | 5 |
| Hiking tours | 4 | 2 | | | | 1 | | 7 |
| Riding tours | 1 | 1 | | | | 1 | 1 | 4 |
| Museum/information | | | 1 | 1 | 2 | 2 | | 6 |
| Other (shop/rafting/biking) | | | | 1 | | 3 | | 4 |
| Total | 17 | 6 | 6 | 2 | 8 | 19 | 7 | 65 |

Three of the municipalities (Skaftárhreppur, Þingeyjarsveit and Skagafjörður) have suffered from out-migration in the past two decades although they have started to rebound somewhat in the last few years (Figure 4). Bláskógabyggð in the south west of the country has a locational advantage of being approximately only an hour's drive from Reykavik. Its economy is largely based on tourism and greenhouse production. Skaftárhreppur is primarily an agricultural rural area, without an urban center. It benefits from tourism along the south cost, which has become almost a whole year business. Skagafjörður and Þingeyjarsveit are also agricultural areas but they have, in addition, a fishing sector and the latter has a new energy intensive silicone factory, along with being a significant location for whale watching tourism. Skagafjörður is the major center in Iceland for river rafting, although that is only practiced in the summer.

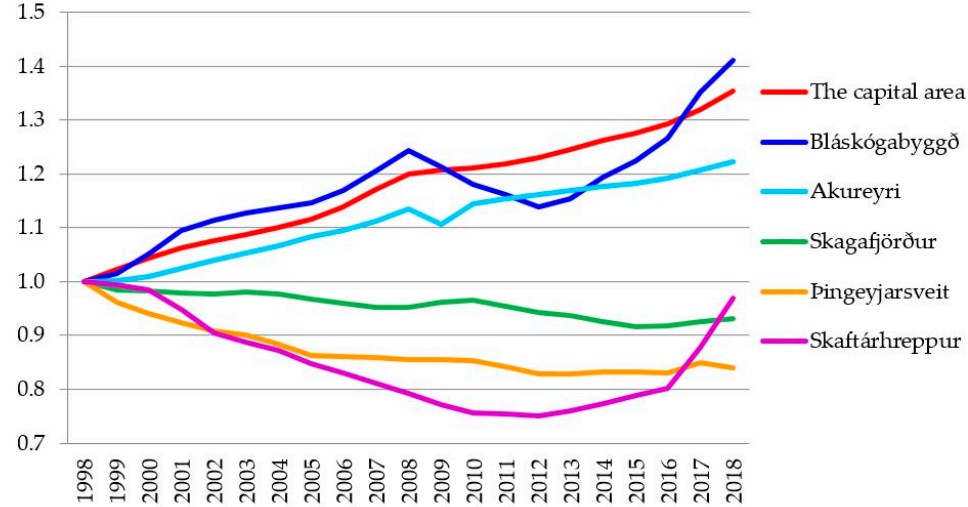

**Figure 4.** Population development in the research areas. Analyzed from Statistics Iceland [65].

## 4. Results

### 4.1. Tourism Operators' Attitudes Towards Power Plants

The results indicated a negative attitude among tourism operators towards the various forms of sustainable power production (hydro, geothermal and wind) and towards power plant constructions

(reservoirs and transmission lines) (Figure 5). The most negative attitudes were towards transmission lines, hydro power plants and reservoirs in the Highlands, and then towards transmission lines in the lowlands. The least negative attitudes were towards wind farms and geothermal power plants in the lowlands, but still over half of the respondents are very negative or somewhat negative towards them. Transmission lines were, in addition, perceived more negatively than reservoirs.

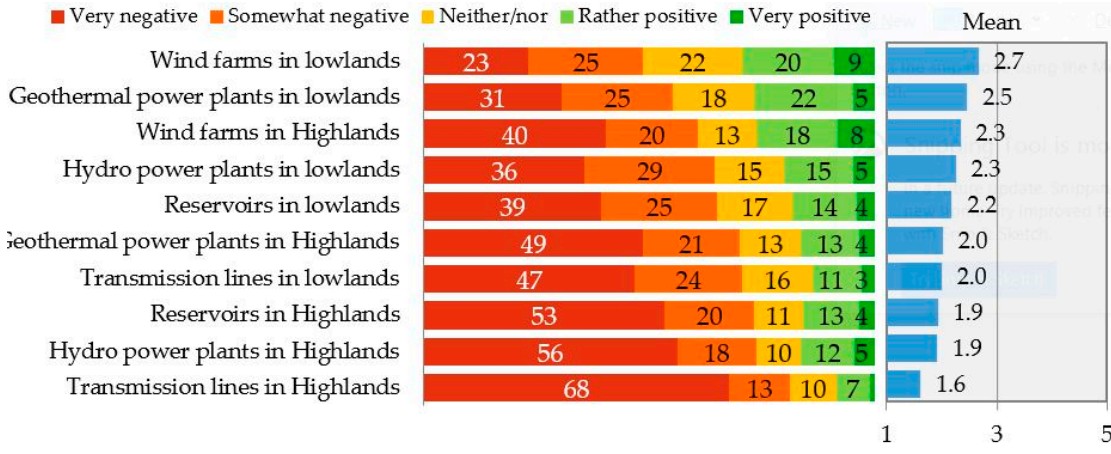

**Figure 5.** Tourism operators' attitudes to the various forms of sustainable power production and power plant constructions.

By comparing the various forms of power production and constructions, and whether the tourism operators prefer their presence in the Highlands or lowlands, it could be seen that respondents were significantly more negative towards all types of power plants in the Highlands than in the lowlands (Table 3). The participants were more negative towards hydro power plants than geothermal power plants in the lowlands, although there was not a significant statistical difference between these two types in the Highlands, where both were considered very negative.

**Table 3.** Tourism operator attitudes towards power plant development and related structures.

| Power Plant Infrastructure | Mean | Std.dev. | *t* | *p* |
|---|---|---|---|---|
| Wind farms in the Highlands * | 2.34 | 1.38 | −3.84 | <0.001 |
| Wind farms in the lowlands | 2.67 | 1.29 | | |
| Geothermal power plants in the Highlands * | 2.02 | 1.22 | −6.19 | <0.001 |
| Geothermal power plants in the lowlands | 2.46 | 1.27 | | |
| Hydro power plants in the Highlands * | 1.92 | 1.25 | −4.91 | <0.001 |
| Hydro power plants in the lowlands | 2.25 | 1.23 | | |
| Reservoirs in the Highlands * | 1.94 | 1.21 | −3.84 | <0.001 |
| Reservoirs in the lowlands | 2.19 | 1.22 | | |
| Transmission lines in the Highlands * | 1.62 | 1.03 | −5.75 | <0.001 |
| Transmission lines in the lowlands | 2.00 | 1.16 | | |
| Hydro power plants in the lowlands * | 2.25 | 1.23 | −2.95 | 0.004 |
| Geothermal power plants in the lowlands | 2.46 | 1.27 | | |
| Hydro power plants in the Highlands | 1.92 | 1.25 | −1.88 | 0.061 |
| Geothermal power plants in the Highlands | 2.02 | 1.22 | | |

Means based on a 5 point Likert-scale where 1 = Very negative → 5 = Very positive (N = 216). * Significant difference between effects of power infrastructure in the lowlands and Highlands at the 0.05 level.

The respondents were asked as to whether the power plants that have already been built had had a negative impact on their business. Just over 40% responded that the impacts were either very negative or negative, although 38.5% stated there was no impact. The respondents were also asked

to evaluate the impact that each of the 26 power plant proposals in the Master Plan project would have on both their company and on the tourism industry as a whole (Figure 6). The most negative effect was considered to be at a pristine salmon catching river (Stóra-Laxá) in the Highlands and all the hydro power plant proposals in the northern Highlands (on the Skjálfandafljót river in the north east and in Skagafjörður in the north west). The two proposals in the centre of the Central Highlands (Skrokkölduvirkjun and Hágönguvirkjun) were also considered negative, as well as the ones in the south besides the river Hólmsá. The least negative attitudes were towards the two proposed windfarms (Búrfellslundur and Blöndulundur) and some of the hydro power plant proposals in Þjórsá, a river in the southern lowlands which is already harnessed quite extensively for power production. The geothermal power plants proposals on the Reykjanes peninsula (a lowland area) were also considered to have somewhat less negative impact than some others.

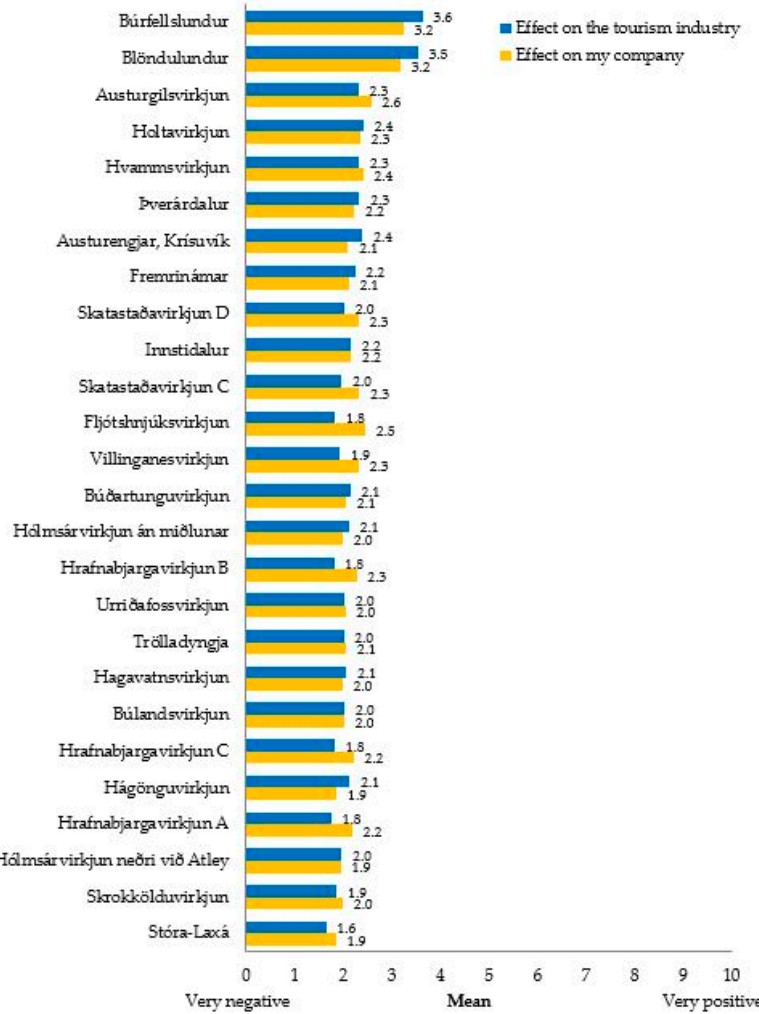

**Figure 6.** Tourism operators' evaluation on the effect each of 26 power plant proposals in the Master Plan project.

In 15 out of 26 instances there was a statistically significant difference regarding how the tourism operators evaluated the effect on their own company and on the industry as a whole. In all instances they evaluated the effects on their own company to be more negative than on the industry overall (Table 4).

**Table 4.** Comparison of tourism operators' evaluation on the effect of each of the 26 power plant proposals in the Master Plan project on the tourism industry and on their company.

| Power Plant Proposals | N | Effect on the Tourism Industry | | Effects on the Company | | Paired *t*-test | |
|---|---|---|---|---|---|---|---|
| | | Mean | Stdev. | Mean | Stdev. | *t*-value | **P** |
| Austurengjar, Krísuvík | 110 | 2.09 | 2.843 | 2.12 | 2.920 | −0.222 | 0.825 |
| Austurgilsvirkjun | 100 | 2.28 | 3.105 | 2.51 | 3.017 | −1.415 | 0.160 |
| Blöndulundur | 123 | 3.24 | 3.207 | 3.16 | 3.006 | 0.596 | 0.552 |
| Búðartunguvirkjun * | 110 | 1.77 | 2.681 | 2.05 | 2.759 | −2.061 | 0.042 |
| Búlandsvirkjun | 112 | 1.76 | 2.715 | 2.02 | 2.774 | −1.931 | 0.056 |
| Búrfellslundur | 124 | 3.37 | 3.267 | 3.23 | 3.047 | 1.003 | 0.318 |
| Fljótshnjúksvirkjun * | 104 | 1.77 | 2.706 | 2.30 | 2.821 | −3.226 | 0.002 |
| Fremrinámar | 111 | 2.05 | 2.857 | 2.15 | 2.832 | −0.707 | 0.481 |
| Hagavatnsvirkjun | 114 | 1.68 | 2.799 | 1.98 | 2.813 | −2.277 | 0.025 |
| Hágönguvirkjun | 112 | 1.90 | 2.959 | 1.88 | 2.802 | 0.126 | 0.900 |
| Holtavirkjun * | 110 | 2.05 | 2.939 | 2.35 | 2.894 | −2.482 | 0.015 |
| Hólmsárvirkjun án miðlunar * | 115 | 1.75 | 2.642 | 1.97 | 2.595 | −2.077 | 0.040 |
| Hólmsárvirkjun neðri við Atley * | 115 | 1.63 | 2.556 | 1.92 | 2.603 | −2.909 | 0.004 |
| Hrafnabjargavirkjun A * | 108 | 1.69 | 2.670 | 2.14 | 2.722 | −3.323 | 0.001 |
| Hrafnabjargavirkjun B * | 108 | 1.75 | 2.697 | 2.23 | 2.801 | −3.726 | <0.001 |
| Hrafnabjargavirkjun C * | 107 | 1.73 | 2.662 | 2.17 | 2.752 | −3.445 | 0.001 |
| Hvammsvirkjun * | 113 | 2.01 | 2.899 | 2.41 | 2.887 | −3.367 | 0.001 |
| Innstidalur | 106 | 2.03 | 2.913 | 2.06 | 2.797 | −0.195 | 0.846 |
| Skatastaðavirkjun C * | 105 | 1.90 | 2.765 | 2.28 | 2.765 | −2.663 | 0.009 |
| Skatastaðavirkjun D * | 105 | 1.88 | 2.706 | 2.26 | 2.746 | −2.692 | 0.008 |
| Skrokkölduvirkjun* | 115 | 1.57 | 2.534 | 1.94 | 2.706 | −3.146 | 0.002 |
| Stóra-Laxá * | 109 | 1.51 | 2.591 | 1.88 | 2.724 | −3.154 | 0.002 |
| Trölladyngja | 109 | 1.85 | 2.520 | 2.01 | 2.713 | −1.069 | 0.287 |
| Urriðafossvirkjun * | 113 | 1.68 | 2.756 | 2.01 | 2.773 | −2.941 | 0.004 |
| Villinganesvirkjun * | 106 | 1.87 | 2.757 | 2.25 | 2.831 | −2.897 | 0.005 |
| Þverárdalur | 106 | 2.20 | 2.919 | 2.27 | 2.965 | −0.508 | 0.612 |

\* Significant difference between effects on the tourism industry and effects on the company at the 0.05 level.

### 4.2. Effects of Power Plant Development on Tourists' Experience

The results of the interviews reinforced the survey finding that the majority of industry respondents believed that power production and nature-based tourism are generally conflicting forms of land use. Power plant constructions were regarded as spoiling the landscape and negatively affecting the experience of tourists by reducing the intrinsic qualities of nature and wilderness. It was believed that this could then lead to flow-on long-term negative impact for the tourism industry in terms of destination attractiveness. As an example of this view, a tourism operator in Reykjavík said:

> 'We are mainly selling access to the nature and selling beautiful nature and every man-made structure that comes in the highland obviously ruins the experience for our people. I believe that power plants have a negative impact, first of all on nature, the wilderness and the outback sense and continuous land and landscape, it would definitely have a negative impact. If 80% of those who come to Iceland are coming because of the nature and sense of wilderness and the stillness, if we are then systematically against this experience by putting up power plants in these areas, then I believe it has a negative impact. It would in the long term, minimize the arrival of tourists to the country.'

Many of the tourism operators emphasized the importance of the Highlands and wilderness, which was 'something which made Iceland unique'. They thought that if more power plants would be built in the Highlands, the wilderness would be more difficult to sell as a tourism product and it would reduce the size of the area that has high natural values in which they can operate. As an example, a person who organizes hiking tours in the Highlands stated:

> 'With Skrokkalda and Hágöngur (two proposals, a hydro and geothermal, in the center of the Highlands) we are ruining a potentially great area for outdoor activities that have not yet

been discovered. If they build a power plant, then I would not organize more tours there again. It is just not possible. It will have gone under water the paths that I use, apart from it being turned into an industrial area and it is just completely uninteresting.'

These two proposals in the Centre of the Highlands would both have an indirect impact by reducing the naturalness of the area and a direct impact on the business, since a hiking route that is currently used would disappear under a reservoir for hydro-electricity. Some respondents note that existing power plants have already had a negative impact on tourism and outdoor recreation in some areas, as well as affecting particular tour offerings.

'The Hengill area (where there is a geothermal power plant) I feel is a warning of things we should avoid. It has an incredibly negative visual impact and the argument that instead you can go and look at some beautiful 3D shows in some powerhouse is off course a completely absurd response to destruction of nature . . . . I think it's a complete catastrophe. It's been implemented in such haste and little thought given to visual impact, just a mass of shiny masts and lines that lie there and cut the land criss-crossed.'

In addition, some respondents were concerned about the wider impacts of energy developments, especially via their effect on Iceland's destination image:

'It destroys Iceland's image and compromises the stakes for tourism this heavy industry, we should rather develop tourism with care and relax with the energy ... aggressiveness of the energy industry.'

Some respondents believed that energy developments would affect their capacity to sell their tourism businesses in the future. One respondent who provides tourism accommodation made the following comment:

'I only know that I want to sell my business and I know it is hard, because power lines all around this area here will destroy so much, that I cannot sell. Nobody will buy here... maybe I'm a little bit old fashioned and old man for this ... because 80% of our visitors to Iceland are coming because of nature.'

However, renewable energy production is also regarded by some as a source of pride and as a positive national identity, even in terms of the conversations they have with tourists. One respondent who runs jeep safari tours said:

'When I give a lecture for the guests in my tours, I praise us in Iceland for utilizing green energy. I think we should be proud for producing green energy rather than using coal and such.'

An interviewee in Reykjavík said:

'I don't see any danger in building more power plants . . . not for the biggest majority who comes here . . . like the American public, they are not concerned about if the water in one small unknown waterfall, has reduced due to some reservoir.'

Some of the interviewees pointed out the attraction of the visitors' centers that have been built at some of the power plants. Furthermore, the attraction of small private power plants that some farmers have made, both hydro and geothermal, was mentioned by some respondents who also suggested that they should be employed more as tourism attractions by advertising them more. However, interviewees did agree that visiting a power plant was 'not the purpose of any tour, except perhaps some minority niche market but they were rather some kind of an add-on/icing on the cake'.

With respect to the relative acceptability of some types of renewable energy developments, some of the interviewees were very negative towards wind farm proposals as they would have a 'devastating effect on the landscape' and considered that, instead, Iceland should 'continue to utilize the unharnessed rivers and develop hydro power plants in order to preserve the landscape'.

### 4.3. Alternatives for Regional Development or Tourism and Rural Development

Most of the interviewees based in rural areas were concerned about the extensive out-migration that has characterized their regional development in recent decades. Amongst the possible causes that were mentioned were changes in the labor market, e.g., fewer jobs in agriculture and the fishing industry, but also altered demands regarding entertainment and service. When an interviewee in Skaftárhreppur was asked whether he considered out-migration to be a cause for concern in the municipality he responded:

'If you had asked me this four years ago I would have said yes, however, I would say no today since tourism is just growing so much ... as a matter of fact we need people to fill full-time and seasonal positions to solve this.'

Most of the interviewees considered that tourism could be credited for the reversal in regional development. An example of this attitude can be seen in the comments of an interviewee in South Iceland who said:

'Certain municipalities would be completely ... well, they would be kind of deserted if they were not tourist places. This is changing, e.g., in Mývatnssveit or Vík í Mýrdal, Öræfasveit ... The out-migration has slowed down because in the countryside guesthouses are popping up here and there and young people who had moved to the concrete jungle now see opportunities to do something.'

Several interviewees felt positive about the future, and the growth in tourism was primarily responsible for their optimism:

'Tourism has been the principal factor in reducing out-migration and we have even begun to see signs that ... young people are starting to arrive just because of tourism, in Mývatnssveit in particular and now also in Laugar.'

The effects of energy production, as well as plans for heavy industry, have been no less positive than those of tourism in Suður-Þingeyjarsýsla. Nevertheless, the majority of the interviewees believed that tourism was the industry with the highest possibilities for generating revenue and strengthening rural communities in the long term. They considered that the sector could expand even further if it was developed appropriately. In this context the importance of rural areas was at the forefront of their thinking because of the proximity to nature—the country's main attraction. Townships bordering the Highlands were regarded as also benefitting from 'having untouched areas ... something that becomes more and more valuable for us to have'. Many tourism service providers pointed out that the Golden Circle (Þingvellir, Gullfoss and Geysir) and the South Coast were verging on reaching their carrying capacity and suggested that new areas would then be needed for tourists to visit. Therefore, North Iceland was regarded as being crucial: 'And when things get more crowded ... in the Golden Circle and the South, and definitely if there is a direct flight to the North, this area would see a huge increase'.

One interviewee in North Iceland reminisced about when he 'founded this tiny company five years ago and today it has 70 people working full-time and after or by the end of the year there will be about 140 people working here full-time'. He continued:

'The great growth that has taken place in the tourism sector in recent years has made it possible for people to continue living in the same place and the young people that had left for the concrete jungle have started to return, which is wonderful to witness. So I would certainly say that tourism has strengthened the rural communities.'

However, it was also pointed out that many jobs in tourism are low-paying: 'This is a low-paying area, that is the big problem if people are after a serious salary, and the driving force behind tourism and gardening and all the manual labor are now foreigners'. However, not everyone agreed that jobs in tourism were low-paying. The owner of an activity company in the North had this to say on the subject:

'We have a group of well-educated people who take care of the office and service work here in the company and then we have highly trained guides who are not low-income individuals so all this talk about e.g., that all tourism jobs are low-paying … it just does not apply to this company.'

Moreover, most interviewees considered that tourism was a much better option than energy production when it came to the creation of new jobs, 'except during the construction period'. An interviewee in the South compared the creation of employment due to energy production with work created by tourism:

' … the Þjórsá area, there are not many who work there full-time … maybe 10 people … Of course, a power plant development would create an enormous amount of work within the municipality but nota bene [note well], for what, four to six years during the construction period? … So, the full-time positions that will be left for the future are possibly limited, one to two posts. Meanwhile a prosperous hotel in business has around 30-40 employees.'

However, one of the positive aspects of the economics of energy production, even though they were perceived as limited, was regarded as being their certainty:

'I know that people here are looking towards the fact that at least this is a secure income, some x millions a year and then it is compared to running the kindergarten or the elementary school and I understand this comparison when they are comparing.'

Many respondents did comment on the positive employment and economic contributions from energy production during the construction stage but emphasized that the employment returns could not be sustained in the long-term. The distribution of the economic returns from energy production and high energy consuming manufacturing was also criticized.

'We haven't been able to gain enough from power production in Iceland, until now, for some reason. We are producing 90% of the electricity for others than ourselves in reality, for Icelandic households. And the fact that these milking cows don't yield more profit for us is really remarkable you know. There is no distribution of capital around it … really.'

Interestingly, the economic returns from tourism were regarded as distributing 'the capital a bit more evenly.' Although the dominance of the capital city region was still noted: 'Don't become a slave of the tourist offices or those in Reykjavik that are making all the money of it by just selling tours.' Some respondents suggested that tourism was also subject to criticism for historically not providing significant enough returns, while some operators also sought to argue that the perception of tourism as contributing only low paid jobs was wrong. Furthermore, tourism was regarded as helping to directly provide for infrastructure that is used by the wider population.

'Tourism is much better, it builds systems and infrastructure for a society that is difficult to maintain for only 300,000 people so with tourism there are so many things … like the swimming pools in Reykjavik. To get one million tourists extra to come and pay for the entrance, it makes the expenses and offers the possibility to maintain and add to the service for the locals and those that live here during the whole year.'

Nevertheless, tourism was also regarded as contributing to housing displacement and a growth in migrant labor:

'They are having problems with people and all houses that has become available has been bought up by the tour operators for staff and they are leasing their dwellings and everything possible for people to live in and then there is so much import of labor.'

Yet, tourism is still regarded as providing more 'opportunities' and contributing more to halting rural depopulation than energy production, especially in some more peripheral areas. As one respondent stated, 'I don't know if it's debatable but we would be bankrupt if we didn't have tourism, it's quite simple.'

### 4.4. Coexistence and Regional Development

A number of respondents commented that growth in energy production and tourism could potentially co-exist in the future. However, it was regarded as requiring improved spatial planning and management, as well as discussions between the two industries so they could coexist. Many of the interviewees mentioned the improved roads and accessibility which accompanies power plant development as the most important positive effect they have on tourism. Still, when roads are constructed the tourism sector believed that they should be consulted in order for both industries to benefit from the developments. A comment from a tourism entrepreneur in the north, near Myvatn, shows an example of the problem:

> 'The tourism industry can make use of the good roads that often accompany the construction of power plants but that is not always the case though . . . We have a considerable amount of Northern lights tours from Lake Myvatn area that going somewhere in the vicinity of Krafla (an existing geothermal power plant), is out of the question, there is an enormous light pollution from Krafla. If Þeistareykir (a new geothermal power plant nearby), are to be designed with as much light pollution then Þeistareykir will not be, despite a straight and wide road and in fact a good construction for tourism, then it will not be a place for Northern light viewers.'

The potential of developing some tourism to energy production sites was mentioned: 'The geothermal fumes and the hot water fascinate a lot of people, people are very excited about the power production and how the hot water is used, I think there are a lot of people visiting Nesjavellir and Hellisheiði to see it.' Nevertheless, it is also apparent that some respondents believe that there is an overly narrow focus when considering regional development alternatives.

> 'There are so many other things possible than aluminum factories, the huge, what do we say, the factories that need the huge power. They could do something smaller, and what I believe they should do that ... look back a bit, think about the film making, the music, all these small things, all of a sudden they pass out, on a world scale, these people are doing so good job.'

Meanwhile other respondents suggested that there was not enough connection made between tourism and domestic agricultural production:

> 'Increasing tourism is positive for agriculture because you have to increase production to feed these 2 million tourists that will arrive here.'

### 4.5. The Future

With respect to the selection of the best resource use options for the future, many respondents argued that, over the long-term, tourism would provide the best economic benefits for the country:

> 'This (tourism) yields the highest profits, this yields a lot more in the next thousand years unharnessed rather than harnessed for electricity ... it is quite obvious.'

Many respondents were extremely positive about the future of tourism arrivals to Iceland and believed that many more could be accommodated, especially if the landscape impacts of energy production could be minimized:

> 'When we are not scaring people away with electric cords and power plants, pipelines, pipes and cords or some horribleness ... we should just set the mark at 12 million tourists and prepare for that and on 12 million tourists every Icelander can live a fine life.'

Many of the concerns over the expansion of energy production and its environmental impacts were directly related to the extent to which energy production potentially foreclosed the long-term economic returns from nature-based tourism.

'I think it's important for us as a nation first of all to start admitting and realize how important nature is for so many reasons. The nature is the biggest attraction and yields the most revenue for society but apart from that I think for the future to have such a country and nature and obtain or to have the idea that this is something we want to preserve and protect for ourselves and upcoming generations because this will continue to become increasingly important worldwide. And power plants even though they yield some financial revenue for a limited amount of time then the time will pass in 30 or 50 years and we are left with irreversible construction. We lack this idea or possibility for so many of us in this society to say, I have something valuable and I want to keep it, take care of it and return it ahead.'

Some of the interviewees raised some concerns regarding the fast rate of growth in numbers of international arrivals to the country and the need to better manage tourism no less than energy related developments.

'Both sectors are moving too fast in my opinion ... On the other hand, I believe the management of tourists is very bad in Iceland and no thought in how we are doing it ... Off course we need to address it and organize it better ... But there is no doubt in my mind that 5 million tourists leave a less impact that Hálslón reservoir ... you know what I mean?'

## 5. Discussion

This paper has examined the perspective of tourism operators in Iceland with respect to renewable energy production, geothermal, hydro-electricity and wind power, and their perception of consequent implications of such development on tourism. This is a major natural resource and sustainable development policy issue in Iceland that has implications not only for tourism actors but, also, on land use and the socio-economic trajectories that municipalities and regions will take into the future [13]. The portrayal of development being a choice between energy production or nature-based tourism is a little simplistic, but it is how many of the respondents perceive the main rural development options for municipalities. While there is a small market for industrial tourism that encourages visits to energy plant and hydro-electric dams, what Frantál and Urbánková [17] refer to as 'energy tourism', most operators, as well as most research suggest that power plants and their following infrastructure have a significant negative impact on landscape perceptions and wilderness qualities [19,53]. Although, perhaps somewhat paradoxically, at the national level the renewable energy developments may not be completely negative and may possibly serve in the short-term to reinforce Iceland's international image of being 'clean and green' [19]. For example, a story on the Hellisheiði plant in The Guardian, featured an interview with Marta Rós Karlsdóttir, managing director of natural resources at ON Power, the publicly owned energy company that runs Hellisheiði and concluded, 'According to Karlsdóttir, a major proportion of the plant's thousands of annual visitors are British schoolchildren, witnessing a vision of clean, sustainable power dramatically different from the murky, fossilized industry they are used to' [102]. Another example of positive relationship between power production and tourism is Reykjanes Geopark, which is one of Iceland's two Geoparks. Geoparks are UNESCO-recognized areas that are of international geological significance where the aim is to combine conservation of unique geological phenomena and community involvement in order to achieve sustainable development. The geothermal energy has been used for power production in Reykjanes Geopark and the Blue Lagoon, one of the most popular tourist destinations in Iceland, is located next to a power plant utilizing its waste water. However, while such sites highly developed sites are significant as part of international tourists' visits to Iceland the majority of the activities outside of the capital region and Keflavík are influenced by their perceptions of high degrees of naturalness.

The Icelandic situation mirrors some of the longstanding observations regarding contested use of wilderness space and understandings of what constitutes good conservation practice and appropriate sustainable development, particularly in peripheral regions in countries such as Australia, Canada and New Zealand [2,6,30]. Tourism is deeply embedded in such contestation as it is often proposed as

an economic alternative to the exploitation of energy resources [5]. However, much of the discussion regarding tourism in wilderness areas often fails to examine the relative perceptions of tourists of development in the landscape. This study therefore makes an important contribution to better understanding how tourism markets perceive energy infrastructure and the longer-term implications for regional economic development [11,13,14].

According to research conducted for Landsvirkjun, the national power company [103] international tourists in Iceland are positive towards renewable energy production in the country and claimed that power plant constructions did not have a negative impact on their experience. The most negative attitude was towards transmission lines, with 18% claiming that they had a rather negative impact on their experience. However, it should be noted that the survey was conducted after tourists had left the areas which are in question and being asked if something which already exists is acceptable there is different to asking what the character of a preferred environment. However, in the longer-term the more energy infrastructure is developed the greater the loss of perceived wilderness values will be and therefore, potentially, the lowering of the quality of the wilderness experience [104]. Although the Highlands have historically been substantially affected by deforestation and overgrazing and other elements of anthropogenic change, it is important to stress that people respond to what they perceive as high naturalness and wilderness values, even though such notions of naturalness are socially constructed and not empirically based on ecological history [83]. Such changes in the perceived naturalness of areas with high wilderness values are likely to have a consequent impact on visitor experiences and potentially visitor numbers, at least in the nature-based tourism market [19] that comprises around nine in every ten visitors [76].

In some ways the framing of Icelandic regional development as energy versus environment revisits many of the dilemmas that the conservation movement has faced since the creation of the Sierra Club and the campaign of John Muir to prevent the damming of Hetch Hetchy at the start of the Twentieth Century [105]. The issue of economic conservation, via renewable energy production, versus wilderness preservation in which economic value is achieved via tourism, has played itself out through much of the peripheral developed world since [2,104]. However, the implications of such development decisions are now much more complex. For example, renewable energy [14–16,33] and nature-based tourism [5,9,10] are individually portrayed as being 'green' and 'sustainable' even though poorly managed growth in energy infrastructure and visitor arrivals will have potentially significant negative environmental impacts, including the perceived quality of the landscape [44,45] and associated visitor experiences [34]. Furthermore, whether it be in the international literature or in the present study, in assessing sustainability tourists do not appear to acknowledge the significant emissions resulting from international travel and the indirect long-term changes to the landscape that will occur as a result of climate change [106,107] or other related long-term impacts of travel such as the introduction of exotic flora [108]. For example, tourism in Iceland accounts for 35% of all carbon emissions, and is responsible for the largest part of emissions, even more than metal production which is the second at 29% [109].

In the Icelandic context, one of the main aims of the strategic regional planning process is the mitigation of differences in living standards and competitiveness of regions by supporting regions with long-term depopulation, unemployment, and a dependence on a single industry. The Regional Plan for 2014–2017 [110] emphasized the importance of the power intensive industry and tourism services alike by emphasizing the importance of road improvements and maintenance. The 2018–2021 plan [111] provides more of an emphasis towards tourism:

- Charging visitors a fee that will be used to develop and maintain tourist destinations and giving municipalities a higher share of tax revenues.
- Have more flight gateways into the country by making Akureyri and Egilsstaðir more attractive with the aim of improving the distribution of visitors around the country.
- Use nature conservation and nature-based tourism to strengthen rural communities.
- Increase knowledge among managers in rural tourism businesses.

- Expand knowledge among those who work infrastructure development in natural areas (e.g., with the aim that design fit well into landscape and supports a positive experience for tourists).
- Develop a destination management plan for each part of the country.
- Innovation in food production.
- Support the development of 'small' power plants, up to 10MW.

The framing of the socio-economic development trajectories for rural areas of Iceland is therefore shifting more towards tourism development with the benefits of energy production seen as being relatively short-term in terms of employment as well as being economically limited. Significantly, while some respondents perceive the resource rent derived from energy infrastructure as being secure over the life of the plant, the economic benefits of tourism are regarded as being better dispersed through communities and regions than those from energy developments and much more substantial with respect to employment over the longer-term. This also reflects some of the policy arguments that have been forwarded in other countries with respect to restricting energy infrastructure in wilderness areas [5,9,10,30]. Nevertheless, in Iceland concerns over housing costs and pressures as a result of the conversion of housing to short-term rentals, often via AirBnB, clearly exist, as do the pressures deriving from having to hire temporary and short-term staff from outside of municipalities, many of which come from Europe for such work, because of the lack of an available local labor force at times of high tourist demand.

Even with a shift in strategic regional planning initiatives towards tourism, the greatest threat to the tourism trajectory, which is based on natural area tourism, from the perspective of industry stakeholders is regarded as being the infrastructure established for energy production in the Highlands, which directly affects the wilderness experience so many are looking for [53–55], together with products such as hiking routes and trails and a dark night sky for observing the northern lights. Limited common ground is perceived as existing between the two sectors except, in some cases, better accessibility to new destinations and visitor centers at some of the power plants. Many respondents mentioned improved management and planning, meaning limiting infrastructure impact on tourism, as a desirable future for rural areas. Most operators also currently do not believe that visitors will adjust to the presence of energy infrastructure over the longer-term, even though there is some evidence to suggest this may be possible, at least for some sections of the tourist market [57].

Iceland's first national planning strategy, the National Planning Strategy 2015–2026, emphasizes the protection of nature and landscape of the Highlands, especially wilderness areas [112]. Any development of power production, tourism, and transport infrastructure should therefore be undertaken with wilderness protection and nature conservation as a major goal. Different levels of accessibility reflect different tourism services, the level of visitation and different land uses. As such, the planning strategy mirrors the essence of the Recreation Opportunity Spectrum (ROS), the commonly referred to framework which involves zoning outdoor recreational areas into classes ranging from developed to undeveloped and identifies opportunities for recreation in each setting [113]. However, while such approaches can help reduce the level of competition between competing land uses through appropriate spatial planning and design strategies and help meet different levels of expectations from different markets, it is likely that some conflicts will remain, especially with respect to visual changes to the landscape.

## 6. Conclusions

In general, the tourism industry and, the communities that have embraced it, reflect the difficulties of long-term economic planning in a 'neoliberal laboratory' for nature and the resource-based sectors [114,115]. Adequate evaluation of the relative environmental, economic and social trade-offs between different development options and their distributional effects has not been fully implemented by the Icelandic government. Perceived market demands, whether tourist or energy have tended to dominate national decision-making although there is increased public concern over the implications of this course of action, especially with respect to the impact tourism has had on housing issues [116].

Such a situation reflects the experiences in the planning of peripheral areas in other developed countries [2,5,9,10] and highlights the importance of an improved understanding of the different roles and influence of institutions, policy actors, and community members in influence the acceptability of energy development and infrastructure [117–119].

The Icelandic situation has the potential to be particularly instructive with respect to issues of energy infrastructure development because of the concerns over the perspective of a temporary non-resident population—the tourist, and their response to sustainable energy development related environmental change. The paper therefore builds on the limited previous research on tourism perceptions of the acceptability of energy infrastructure in areas with high scenic values [19,47,53,55] by examining the attitudes of tourism operators and business people rather than the tourists themselves. This is also significant because the tourism businesses enable the accessibility of many individual tourists to peripheral area locations and determine their movement both within the landscape and in the wider regional context. Operator decisions will therefore ultimately influence where and how much tourist spending will occur and the wider flow in regional economies [9,10].

The long-term implications for regional population and employment trends of the extraction of land-rent from natural resources therefore remains a fraught issue in Iceland, as well as in other locations [9,10], with the danger of lock-in to particular economic and environmental trajectories remaining while the spatial competition between nature-based tourism and energy production remains unresolved. In seeking to manage these issues the country is facing a choice between two different visions of sustainability that have vexed policy makers for over a century. Should the economic justification for conservation be based on the use of renewable energy resources the development of which will fundamentally change the landscape or should it be based on the retention of the landscape in what is perceived as a relatively undisturbed state? Either way, a failure to provide flexible options for the future by effectively managing energy and tourism demands will permanently affect the country's natural resource base and consequent socio-economic and environmental wellbeing.

**Author Contributions:** Writing, review and editing—A.D.S.; C.M.H.

**Funding:** This research was funded by the Ministry for the Environment and Natural Resources and the steering committee for the Icelandic Master Plan for Nature Protection and Energy Utilization.

**Acknowledgments:** We thank the Ministry for the Environment and Natural Resources and the steering committee for the Icelandic Master Plan for Nature Protection and Energy Utilization for financing the data gathering for this research. We also thank Guðmundur Björnsson, Laufey Haraldsdóttir and Georgette Leah Burns for conducting part of the interviews and Anna Mjöll Guðmundsdóttir for transcribing them.

**Conflicts of Interest:** The authors declare no conflict of interest.

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
