# Peer review of "Contested Development Paths and Rural communities: Sustainable Energy or Sustainable Tourism in Iceland?"

_sustainability, doi:10.3390/su11133642_

Round 1

Reviewer 1 Report

This manuscript has the potential to become an interesting paper, but it remains poorly structured and poorly reported in its current version. Besides, it should be clearly stated what aspect of sustainability is being addressed here (i.e. what is the central issue in terms of 'sustainability'?) 

I believe the paper should eventually be published due to the significance of the topic, but some major improvements must be made: 

Main concerns about the work: 

The first major problem is with the method used: in order to 'examine' which option (hydro development or tourism) is better and environmentally sustainable, the authors conducted a survey with tourism stakeholders! This is surprising because naturally the opinion of tourism stakeholders will be against hydro development--owing to conflict of interest. How can this survey be justified as offering adequate insight then? I do believe the findings are relevant, but the scientific soundness of the work needs to be improved; this remains a major issue. 

In order to solve Point 1 above, I thought, after reading the work, that the most appropriate question here is perhaps something like: "What does the perception of the tourism stakeholders inform us about potential environmental damage from hydro-development on landscapes that have significant wilderness/recreation value?" The survey with the tourism stakeholders will then be very much justified, and the results can be described accordingly.

The second major problem of the paper is a lack of contextualization. The authors do not cite adequate literature on environmental sustainability to justify the point of analytical departure and explanation of the results. In fact very little is described on the issue of environmental sustainability of the area under focus (for example. how does hydropower development affect rivers, soil, and physical processes?). There is a lot of relevant literature on alteration of rivercourses for hydro-development and consequent loss of ecological integrity or ecosystem services--some of those must be touched upon. 

In the 'Discussion and Conclusion' section, a clear explanation should be provided on the ecological condition of the landscape in question--currently the results and the discussion remain too narrowly focused on economic aspects.  

I have another additional question on Iceland's environment: it is on how 'wild' it really is. The authors seem to report it as 'pristine' and 'untouched' but there are several studies that document extensive deforestation in Iceland and some studies claim it is among the most deforested of Scandinavian countries: for example see: Edwards, K. J., Lawson, I. T., Erlendsson, E., & Dugmore, A. J. (2005). Landscapes of contrast in Viking Age Iceland and the Faroe Islands. Landscapes, 6(2); Vésteinsson, O., McGovern, T. H., & Keller, C. (2002). Enduring impacts: Social and environmental aspects of Viking age settlement in Iceland and Greenland. Archaeologica Islandica, 2. 

I have some additional minor comments:

P2 Line 69: "Materials Iceland..." is this a typo?  

P 17 Lines 586-587: Please reformulate this sentence in a more lucid manner. 

You refer to 'coexistence' of energy development and tourism but do not provide any reference to UNESCO Geoparks. As I know there are several UNESCO Geoparks in Iceland and that their supposed mandate is to reconcile development, conservation and recreation, I am just curious to know if some of their activities may be of pertinence here. 

Reviewer 2 Report

General comments

The emergence of the idea of sustainable tourism has highlighted the need for a multi-sectoral approach to tourism planning. The focus of this manuscript on potential conflicts between nature-based tourism and energy development in Iceland promises to contribute to the existing literature on sustainable tourism. However, the research problems that inform this manuscript could be clarified further, and more specific research questions could be provided. The methods section also requires a bit more measurement details, as well as details on data analysis. The concluding section of the manuscript could also be enriched with policy recommendations for reconciling the renewable energy versus sustainable tourism divide in Iceland.

Introduction

In the introduction of the manuscript, the authors highlight the potential challenges posed by differences in understanding and perspectives among stakeholders with interest in sustainable development, renewable energy, and sustainable tourism. Overall, this section introduces several concepts, none of which is actually explained. As such, the research problem and knowledge gaps that justify the need for this study do not appear to have been adequately established. Although the need for multi-sectoral planning is now widely recognized across all sectors, its implementation is limited. Hence, there’s a clear need for this study. However, to frame the problem in an interesting manner from a research perspective, it might be a good idea to choose a specific sector, e.g. sustainable tourism. The authors could talk about sustainable development and its relationship with sustainable tourism and nature-based tourism, highlight the need for cross-sectoral planning in sustainable tourism processes and some of the challenges entailed in it. The specific case of the conflict between sustainable tourism and the energy sector in Iceland could then be used as a case study to further explore these issues. Currently, the meaning and relationships among renewable energy and all the other concepts that are used in this section remain unclear. Also, more specific research questions are needed at the end of the introduction to clarify the focus of the manuscript.

Study context

The second section of the manuscript provides a detailed description of the context of the study. This section offers more clarity on the relationships between tourism and the energy sector in Iceland.

Methods

The methods section is also very detailed and the methods used seem appropriate. However, the absence of specific research questions and/or objectives at the beginning of the study makes it difficult to fully determine the appropriateness of the methods. Also, measurement details in the questionnaire that was administered in the online survey need to be provided. What constructs was the questionnaire designed to capture, and how were they operationalized? Similarly, sample questions that were asked during the interviews could be helpful. Finally, the methods section does not provide a description of the analysis of the qualitative and quantitative data that were generated from the study. How were the data analyzed to address the research questions?

Results

The results section seems well-organized and contains a substantial amount of qualitative and quantitative data. The only problem here is that the reader has no clear idea about the exact research question(s) the data are intended to address.

Discussion and conclusion

In this last section of the manuscript, the authors highlight the findings on the perception of potential conflicts between energy infrastructure development and wilderness values in nature-based tourism systems in Iceland.  However, the authors do not suggest any specific measures for resolving these conflicts. For instance, from a recreation and tourism planning standpoint, the recreation/tourism opportunity spectrum is commonly used as a tool for zoning recreational settings and also for analyzing the impacts of proposed policies on existing recreational opportunities. In this regard, one could ask if some forms of tourism will be more compatible with energy development, and if so, whether a diversification of Iceland’s tourism system from a predominantly nature-based one to other forms that could also accommodate energy development to a greater extent could be a feasible option. Framing the research problem from a specific sectoral perspective at the beginning of the manuscript could make it easier to draw from that sector in search of potential solutions to these conflicts.

Author Response

Response to Reviewer 2 Comments

We thank the reviewer for very good and constructive comments which we feel have really improved the quality of the paper.

… the research problems that inform this manuscript could be clarified further, and more specific research questions could be provided.

Introduction…Overall, this section introduces several concepts, none of which is actually explained. As such, the research problem and knowledge gaps that justify the need for this study do not appear to have been adequately established. However, to frame the problem in an interesting manner from a research perspective, it might be a good idea to choose a specific sector, e.g. sustainable tourism. The authors could talk about sustainable development and its relationship with sustainable tourism and nature-based tourism, highlight the need for cross-sectoral planning in sustainable tourism processes and some of the challenges entailed in it. The specific case of the conflict between sustainable tourism and the energy sector in Iceland could then be used as a case study to further explore these issues. Currently, the meaning and relationships among renewable energy and all the other concepts that are used in this section remain unclear. Also, are needed at the end of the introduction to clarify the focus of the manuscript.

·         We have added the following text and research questions:

Iceland has the benefits of growing tourism industry as well as various options for producing renewable energy. So far, several hydro-electric and geothermal power plants have been built and more are under consideration. In addition, the development of the first wind farms in the country are now being discussed. However, studies among tourists at natural area destinations in Iceland show that tourists are negative towards proposed renewable power plant development in natural areas (Burns & Haraldsdóttir, 2018; Stefánsson, Sæþórsdóttir, & Hall, 2017; Sæþórsdóttir, 2010; Sæþórsdóttir, Ólafsdóttir, & Smith, 2017). On the other hand, a study by Sæþórsdóttir and Hall (2018) shows that where a hydro power plant infrastructure has been developed tourists' perception of the naturalness of the area in question has not been severely affected.

When new power plants are built in natural areas new roads are built and improvements made to existing roading. Therefore, new areas become more accessible, which, sometimes lead to increased number of visitors although the experience of the place and the market group that is attracted might be different from what existed previously (Sæþórsdóttir & Hall, 2018; Sæþórsdóttir & Ólafsson, 2010). Consequently, road construction and accessibility have a major effect on how a tourist destination develops, its characteristics, what type of tourists visit the area and to what extent the area is visited.

Based on surveys and interviews with tourism operators the aim of this paper is to examine:

-        How tourism operators perceive the ideas of further development of power plants in Icelandic nature.

-        What impact tourism operators think proposed power plants have on the tourism industry in Iceland.

-        If tourism operators see new market opportunities or a loss of opportunities due to power plant development.

-        If tourism operators perceive that tourism and power production can coexist.

-        If tourism operators perceive tourism or power plants as a better alternative for regional development, or if conflicts foreseeable.

The research fills an existing gap in knowledge in this field, that is the view of the tourism industry on energy developments. That knowledge is of vital importance as the opinion, beliefs and perceptions of the sector influences their actual behavior. It of course not to say that the business is right regarding what will happen to the operation or to tourism as a whole if a power plant is constructed, but beliefs do influence actions and decision-making (Shepherd et al., 2015). If for example an operator in a rural area does not think that their market could coexist with power plant and there it a strong possibility for power plant development they might not spend time or money marketing a tour into the area or may even operate elsewhere. Alternatively, the development of a power plant may be regarded as a potential opportunity to develop new product offerings and develop new markets given increased accessibility and new attractions (Frantál & Urbánková, 2017).

In Iceland such information is of special importance as the tourism industry has become the largest export sector, while the government also seeks to identify the best economic and environmental conservation strategies for areas that are currently perceived as having high natural values.

The methods section also requires a bit more measurement details, as well as details on data analysis….
The methods section is also very detailed and the methods used seem appropriate. However, the absence of specific research questions and/or objectives at the beginning of the study makes it difficult to fully determine the appropriateness of the methods. Also, measurement details in the questionnaire that was administered in the online survey need to be provided. What constructs was the questionnaire designed to capture, and how were they operationalized? Similarly, sample questions that were asked during the interviews could be helpful. Finally, the methods section does not provide a description of the analysis of the qualitative and quantitative data that were generated from the study. How were the data analyzed to address the research questions?

·         We have outlined better regarding measurement details and how data were analyzed by adding:

It contained 21 question that can be grouped into:

•         Their type of business and location of where they take tourists/run their business.

-        What kind of tourism services do you provide?

-        Where does most of your business take place?

-        How many employees work at the company?

-        How many years have you been in business?

•         The effect of existing power plants on the tourism industry.

-        Have the existing power plants had an impact on your business or the way you run it? What kind of impact? Has it been good or bad?

•         Attitudes towards the various types of power plants (hydro, geothermal wind) and related structures as well as their location (Highland versus Lowland) and their further development.

-        Please state how positive or negative your attitude is to the following:

-        Hydro power plants in the Highlands

-        Hydro power plants in the lowlands etc.

-        Further development of hydro power plants in the Highlands

-        Further development of hydro power plants in the lowlands etc.

•         Attitudes towards the various 26 power plant proposals. Here the respondents could open a link at the webpage of the National Energy Authority with a brief description of each of the 26 proposals.

How would you rate the following power plant proposal (on the scale 0-10) regarding how good or bad you think it has on i) the tourism industry on your company and in Iceland).

In total 355 opened the survey, 259 started and 156 finished making a 15.8% response rate. It took those who finished on average about 14 minutes to fill it out.

In the analyses both descriptive statistics were used as well as in the Likert scale questions the means were calculated, and comparison made with independent t-tests to discover whether there was a statistically significant difference. In the following analyses, a significance level of 0.05 is used, i.e. if p < 0.05 it is concluded that statistically significant differences exist.

·         Regarding the semi-structured interviews we added:

The interviews were semi-structured and contained mostly open-ended questions grouped around the following themes and questions:

•         Current utilization of the area (quantity of use and what type of activities/tourists).

-        What kind of business do you run in the area?

-        How many visitors (in your tours, at you place at each time/ in a year)?

-        What do they do while they are here?

-        •         The area´s attraction and its uniqueness as a tourist destination.

-        What kind of tourists do you get and what are their demands?

-        What places are they looking at, what are the most important ones, what is the attraction for travellers?

-        What makes this place special as a destination for travellers?

•         Vision and future possibilities for tourism in the area.

-        What possibilities do you see (in the area) for the future?

-        How do you see the tourism industry developing in the next years?

-        Could the area be used more by the tourism industry? How?

-        Could more travellers come into the area? Can the area tolerate more travellers? Why?

-        What kind of infrastructure is suitable in this area?

-        What do you think future travellers would prefer to have here regarding access and infrastructure (accommodation, restaurants/catering, activities?)

•         Attitudes towards power plant proposals in the operating area and possible influence on tourism.

-        Are you familiar with the power plant proposals? (explain shortly if they are not)

-        What is your opinion on each of them?

-        Would the power plants have impact on you/your company? If yes, how?

-        What impact would these power plants have on tourism? (your own business, others, travellers experience)?

-        How do you think tourism would develop with the power plants?

-        How do you think tourism would develop without the power plants?

-        What do you prefer – why, argue for it?

•         Tourism and/or power production as a solution/economic trajectory for the municipality and national interests.

-        Which do you consider the preferred alternative for Iceland: further development of tourism or energy production? What about for regional development within the municipality? Alternatively, can they coincide?

The interviews reflect on individuals’ version of “truth” (Hannam & Knox, 2005) an the analysis of the interviews is built on a standpoint that beliefs about “nature” are not fixed ontological properties but a socially produced “reality” (Demeritt, 2002; Penman, 2001).

Results

The results section seems well-organized and contains a substantial amount of qualitative and quantitative data. The only problem here is that the reader has no clear idea about the exact research question(s) the data are intended to address.

·         We think this should be clear now since we have added the research questions.

The concluding section of the manuscript could also be enriched with policy recommendations for reconciling the renewable energy versus sustainable tourism divide in Iceland.

Discussion and conclusion

In this last section of the manuscript, the authors highlight the findings on the perception of potential conflicts between energy infrastructure development and wilderness values in nature-based tourism systems in Iceland.  However, the authors do not suggest any specific measures for resolving these conflicts. For instance, from a recreation and tourism planning standpoint, the recreation/tourism opportunity spectrum is commonly used as a tool for zoning recreational settings and also for analyzing the impacts of proposed policies on existing recreational opportunities. In this regard, one could ask if some forms of tourism will be more compatible with energy development, and if so, whether a diversification of Iceland’s tourism system from a predominantly nature-based one to other forms that could also accommodate energy development to a greater extent could be a feasible option. Framing the research problem from a specific sectoral perspective at the beginning of the manuscript could make it easier to draw from that sector in search of potential solutions to these conflicts.

·         We have added the following:

Iceland’s first national planning strategy, i.e. the National Planning Strategy 2015-2026 emphasis the protection of nature and landscape of the Highlands, especially wilderness areas. Any development of power production, tourism, and transport infrastructure should be undertaken with primal concern for wilderness protection and sustainability. Different accessibility is furthermore used to segregate the various level of tourism services as well as the various use levels and other land use (National Planning Agency, 2016). Thereby it mirrors in a way the essence of the Recreation Opportunity Spectrum (ROS), the commonly referred to framework which involves zoning outdoor recreational areas into classes ranging from developed to undeveloped and identifies opportunities for recreation in each setting (e.g. Eagles, McCool, & Haynes, 2002).

·         We have had language and style re-checked.

Reviewer 3 Report

The manuscript is to focus on the attitudes of Icelandic tourism operators towards power production and proposed power plants. Mixed method has been adopted to glean data to explain the attitude of tourism operator. Generally speaking, the manuscript is well written and organised. However, the authors should have some minor revision to clarify some parts of the manuscript.

It is interested to know that why tourism operators are so important to understand their view on proposed power plants? most studies in literature will take local residents into consideration on such development. Authors should further justify why tourism operators have been selected for this study in the introduction section.

Authors should also explore how the results of their study could inform relevant policy.

Authors need to describe and clarify how are there questions (21 questions) in the questionnaire designed? what are they? and why are they included? 

the questionnaire can be attached as an appendix to allow more understanding on the research instrument.

Author Response

Response to Reviewer 3 Comments

We thank the reviewer for very good and constructive comments which we feel have really improved the quality of the paper.

It is interested to know that why tourism operators are so important to understand their view on proposed power plants? most studies in literature will take local residents into consideration on such development. Authors should further justify why tourism operators have been selected for this study in the introduction section.

We have added into the introduction why our study      focuses on the tourism industry:

The research fills an existing gap in knowledge in this field, that is the view of the tourism industry on energy developments. That knowledge is of vital importance as the opinion, beliefs and perceptions of the sector influences their actual behavior. It of course not to say that the business is right regarding what will happen to the operation or to tourism as a whole if a power plant is constructed, but beliefs do influence actions and decision-making (Shepherd et al., 2015). If for example an operator in a rural area does not think that their market could coexist with power plant and there it a strong possibility for power plant development they might not spend time or money marketing a tour into the area or may even operate elsewhere. Alternatively, the development of a power plant may be regarded as a potential opportunity to develop new product offerings and develop new markets given increased accessibility and new attractions (Frantál & Urbánková, 2017).

In Iceland such information is of special importance as the tourism industry has become the largest export sector, while the government also seeks to identify the best economic and environmental conservation strategies for areas that are currently perceived as having high natural values.

Authors should also explore how the results of their study could inform relevant policy.

Authors need to describe and clarify how are there questions (21 questions) in the questionnaire designed? what are they? and why are they included? …the questionnaire can be attached as an appendix to allow more understanding on the research instrument.

·         We have outlined better regarding measurement details and how data were analyzed by adding:

It contained 21 question that can be grouped into:

•         Their type of business and location of where they take tourists/run their business.

-        What kind of tourism services do you provide?

-        Where does most of your business take place?

-        How many employees work at the company?

-        How many years have you been in business?

•         The effect of existing power plants on the tourism industry.

-        Have the existing power plants had an impact on your business or the way you run it? What kind of impact? Has it been good or bad?

•         Attitudes towards the various types of power plants (hydro, geothermal wind) and related structures as well as their location (Highland versus Lowland) and their further development.

-        Please state how positive or negative your attitude is to the following:

-        Hydro power plants in the Highlands

-        Hydro power plants in the lowlands etc.

-        Further development of hydro power plants in the Highlands

-        Further development of hydro power plants in the lowlands etc.

•         Attitudes towards the various 26 power plant proposals. Here the respondents could open a link at the webpage of the National Energy Authority with a brief description of each of the 26 proposals.

How would you rate the following power plant proposal (on the scale 0-10) regarding how good or bad you think it has on i) the tourism industry on your company and in Iceland).

In total 355 opened the survey, 259 started and 156 finished making a 15.8% response rate. It took those who finished on average about 14 minutes to fill it out.

In the analyses both descriptive statistics were used as well as in the Likert scale questions the means were calculated, and comparison made with independent t-tests to discover whether there was a statistically significant difference. In the following analyses, a significance level of 0.05 is used, i.e. if p < 0.05 it is concluded that statistically significant differences exist.

·         We have had language and style re-checked.

Reviewer 4 Report

I really appreciate for the authors’ effort to collect data, information on energy situation and tourism businesses. I have read the manuscript with great interest, but still confused to answer myself whether the following drawbacks of the study can be overcome to be published. I see this paper as no more than a comprehensive report on status quo condition of energy development conflicts with tourism among many business sectors without any theoretical implication nor generalizability of the findings.

Critical issues:

- no clear study objectives (research hypothesis)

- no theory based, thus ambiguous methodology

- low (15.8%) response rate

- old data (2015), considering the nature of the conflict

- less effort to explain 'why' questions, rather focus on explaining the situation.

- conclusions were ‘ecologically fallacious’, meaning study objectives and results were not clearly matched with the conclusions.

Thank you very much.

Author Response

Response to Reviewer 4 Comments

We thank the reviewer for very good and constructive comments which we feel have really improved the quality of the paper.

I really appreciate for the authors’ effort to collect data, information on energy situation and tourism businesses. I have read the manuscript with great interest, but still confused to answer myself whether the following drawbacks of the study can be overcome to be published. I see this paper as no more than a comprehensive report on status quo condition of energy development conflicts with tourism among many business sectors without any theoretical implication nor generalizability of the findings.

- no clear study objectives (research hypothesis)

·         We have added the following research questions:

Based on surveys and interviews with tourism operators the aim of this paper is to examine:

-        How tourism operators perceive the ideas of further development of power plants in Icelandic nature.

-        What impact tourism operators think proposed power plants have on the tourism industry in Iceland.

-        If tourism operators see new market opportunities or a loss of opportunities due to power plant development.

-        If tourism operators perceive that tourism and power production can coexist.

-         If tourism operators perceive tourism or power plants as a better alternative for regional development, or if conflicts foreseeable.

- no theory based, thus ambiguous methodology

·         We have attempted to improve this by adding the following with respect to the issue of the effects of renewable energy infrastructure on perceived naturalness:

Some studies (e.g. Smardon & Pasqualetti, 2016; Wolsink, 2007a) have indicated that renewable energy infrastructure reduces the attractiveness of nature-based tourism destinations. According to Fredman and Tyrväinen (2010) the nature based tourism sector generally owes its business to the perceived naturalness of the landscape. Consequently, conflicts can be expected between renewable power production and the nature-based tourism sector. Furthermore, these conflicts are often greater in high-quality natural areas (Nadaï & van der Horst, 2010) than in areas where there are already industrial plant and infrastructure (Devine-Wright & Batel, 2013).

Some renewable energy infrastructure, such as wind turbines are immense constructions and can therefore have a significant impact on the perceived naturalness of a landscape. Several studies (Devine-Wright & Howes, 2010; Frantál & Kunc, 2011; Smardon & Pasqualetti, 2016; Wolsink, 2007b, 2010) point out that the main reason for the disapproval of wind farms is the apprehension that perceived quality of the landscape will diminish. Several studies (Broekel & Alfken, 2015; Devine-Wright, 2009) point out that travelers sometime stop visiting a tourist destination after wind farm construction. On the other hand, a new segment of tourists sometimes starts to visit an area after the development of power plant, as energy infrastructure can be an attraction for some, while associated roading can make access easier (Eltham, Harrison, & Allen, 2008; Frantál & Kunc, 2011). According to Frantál and Urbánková (2017) such ‘energy tourism’ is growing as a product. In addition, economic reasons can also be a factor that affect the attitudes towards power plant development, e.g. if economic benefits are to be expected people might be willing to sacrifice some of the perceived quality of the nature (Agterbosch, Meertens, & Vermeulen, 2009; Mulvaney, Woodson, & Prokopy, 2013a, 2013b).

·         We have also substantially revised the methods section.

- low (15.8%) response rate

- old data (2015), considering the nature of the conflict

- less effort to explain 'why' questions, rather focus on explaining the situation.

- conclusions were ‘ecologically fallacious’, meaning study objectives and results were not clearly matched with the conclusions.

·         The response rate is not out of the ordinary for other similar research conducted in Iceland.

·         We would note that the data was obtained at the very end of 2015. However, even with this being the case we would argue that the data continues to illustrate some of the core issues faced in the decision-making process in the region and the underlying tensions that exist between different notions of sustainability.

·         We have sought to ensure a better match between the study’s objectives, results and conclusions.

·         We have had language and style re-checked.

Round 2

Reviewer 1 Report

I have read through your revised manuscript and I can see that my major concerns have been addressed. I'm happy to refer it to the next stage, but there are a few minor points that I want you to consider: 

Abstract: Could you provide a simple statement at the end on what general lesson for 'sustainability' can be drawn from the work? 

Introduction: It will be better to explicitly mention that the study presents an overall qualitative analysis of stakeholder perceptions/dynamics. This will help contextualize the study in sustainability related literature. 

P 3 Line 149-151: 'It of course not to say...' the sentence here is odd. Please fix it. In addition i think Lines 149-156 may be excised as the statement made here is quite obvious. 

P 19 Lines 844-846: 'Geoparks are UNESCO recognizded areas who are....' : chnage to 'that are' instead of 'who are', and '..on its path to sustainable development'...this clause also reads odd, may be 'in order to achieve sustainable development...' or something like that? 

I can see you have added a note on the Geoparks but my question was only one of curiosity. The note added at the Discussion/conclusion section somehow feels rather detached from the manuscript, and as there is no detailed discussion on the geoparks it may be excised. 

Author Response

Response to Reviewer 1 Comments

We thank the reviewer for the additional comments which we respond to the following way.

Abstract: Could you provide a simple statement at the end on what general lesson for 'sustainability' can be drawn from the work? 

We have added the sentence:  Although both industries claim to contribute to sustainability they utilize the same resources, and land-use conflicts can be expected, illustrating the contestation that can occur between different visions and understandings of sustainability.

Introduction: It will be better to explicitly mention that the study presents an overall qualitative analysis of stakeholder perceptions/dynamics. This will help contextualize the study in sustainability related literature. 

We have changed the sentence to: Based on surveys and interviews with tourism operators the study presents an overall qualitative analysis of stakeholder perceptions.

P 3 Line 149-151: 'It of course not to say...' the sentence here is odd. Please fix it. In addition i think Lines 149-156 may be excised as the statement made here is quite obvious. 

Thank you we have modified this

P 19 Lines 844-846: 'Geoparks are UNESCO recognizded areas who are....' : chnage to 'that are' instead of 'who are', and '..on its path to sustainable development'...this clause also reads odd, may be 'in order to achieve sustainable development...' or something like that? 

I can see you have added a note on the Geoparks but my question was only one of curiosity. The note added at the Discussion/conclusion section somehow feels rather detached from the manuscript, and as there is Michael, I don’t understand what we have to delete. Can you?

We have modified the comments on the Geopark.

Reviewer 2 Report

This revised manuscript largely addresses the substantive shortfalls that were identified in the previous version in terms of the clarification of knowledge gaps, elaboration on the quantitative component of the data collection process, and a discussion of the policy implications of the findings from a recreation planning standpoint. However, the current description of the survey instrument could be enriched by including the response options to the questions, as well as information on the coding of the responses. Also, in the paragraph on data analysis, it will be nice to clarify which of the statistical procedures were used to address which of the research objectives.

Author Response

Response to Reviewer 2 Comments

We thank the reviewer for the additional comments which we respond to the following way.

However, the current description of the survey instrument could be enriched by including the response options to the questions, as well as information on the coding of the responses.

We have added the green marked text:

•    The effect of existing power plants on the tourism industry.

-         Have the existing power plants had an impact on your business or the way you run it? What kind of impact? Has it been good or bad? replies based on a five-point Likert scale, i.e., 1 = very good, 2 = good, 3 = neutral, 4 = bad, 5 = very bad

•    Attitudes towards the various types of power plants (hydro, geothermal wind) and related structures as well as their location (Highland versus Lowland) and their further development.

-         Please state how positive or negative your attitude is to the following:

o    Hydro power plants in the Highlands

o    Hydro power plants in the lowlands etc.

o    Further development of hydro power plants in the Highlands

o    Further development of hydro power plants in the lowlands etc.

The replies were based on a five-point Likert scale, i.e., 1 = very negative, 2 = somewhat negative, 3 = neutral, 4 = somewhat positive, 5 = very positive.

Also, in the paragraph on data analysis, it will be nice to clarify which of the statistical procedures were used to address which of the research objectives. 

We have added:

In order to discover whether there was a statistically significant difference between the tourism operators´ preferences of the various forms of power production in the Highlands or lowlands, independent t-tests were used. Additionally, to compare tourism operators’ evaluations of the effect of each of the 26 power plant proposals in the Master Plan project on the tourism industry and on their company paired t-tests were used.

Reviewer 4 Report

Thank you for your effort to make manuscript more readable.

Author Response

Thank you!